# The Chimeric Antigen Receptor T Cell Target Claudin 6 Is a Marker for Early Organ-Specific Epithelial Progenitors and Is Expressed in Some Pediatric Solid Tumor Entities

**DOI:** 10.3390/cancers17060920

**Published:** 2025-03-07

**Authors:** Larissa Seidmann, Arthur Wingerter, Marie Oliver Metzig, Angelina Bornas, Khalifa El Malki, Arsenij Ustjanzew, Franziska Ortmüller, Yevgeniy Kamyshanskiy, Thomas Kindler, Mark Laible, Xenia Mohr, Nicole Henninger, Alexandra Russo, Olaf Beck, Francesca Alt, Pia Wehling, Wilfried Roth, Claudia Paret, Jörg Faber

**Affiliations:** 1Institute of Pathology, University Medical Center of the Johannes Gutenberg-University Mainz, 55131 Mainz, Germany; 2Helmholtz-Institute for Translational Oncology Mainz (HI-TRON), 55131 Mainz, Germany; 3Department of Pediatric Hematology/Oncology, Center for Pediatric and Adolescent Medicine, University Medical Center of the Johannes Gutenberg-University Mainz, 55131 Mainz, Germany; 4Institute of Medical Biostatistics, Epidemiology and Informatics (IMBEI), University Medical Center of the Johannes Gutenberg-University Mainz, 55131 Mainz, Germany; 5University Cancer Center (UCT), University Medical Center of the Johannes Gutenberg-University Mainz, 55131 Mainz, Germany; 63rd Medical Department, University Medical Center, Johannes Gutenberg University Mainz, 55131 Mainz, Germany; 7TRON-Translational Oncology, University Medical Center, Johannes Gutenberg University Mainz, 55131 Mainz, Germany; 8German Cancer Consortium (DKTK), Site Frankfurt/Mainz, Germany, German Cancer Research Center (DKFZ), 69120 Heidelberg, Germany; 9BioNTech SE, 55131 Mainz, Germany; 10Research Center for Immunotherapy (FZI), 55131 Mainz, Germany

**Keywords:** Claudin 6, CAR-T cells, desmoplastic small round cell tumors, germ cell tumors, Wilms tumors

## Abstract

The oncofetal membrane protein Claudin 6 (CLDN6) is an attractive target for CAR-T cells therapies. Detailed analysis of CLDN6 expression in normal tissues of children is lacking, limiting its use in pediatric trials. Here, we analyzed CLDN6 expression in pediatric solid tumors and normal tissues across four age groups using RNA-sequencing, qRT-PCR, and immunohistochemistry. We found CLDN6 expression in the fetal epithelial cells of several organs, but not in undifferentiated blastemal cells. Postnatal, we detected CLDN6-positive epithelial progenitors only during the first few weeks of life. We found strong and homogeneous CLDN6 expression in desmoplastic small round cell tumors and germ cell tumors. Wilms tumors demonstrated heterogeneous CLDN6 expression, notably absent in the blastemal component. These findings highlight an organ-specific presence of CLDN6-positive epithelial precursors that largely disappear in terminally differentiated epithelia and support CLDN6 as a viable target for pediatric cancer therapies.

## 1. Introduction

The treatment of pediatric cancer patients is a notable success in medicine, with 82% of patients under 15 years of age in Germany surviving their disease for more than 15 years [1]. However, new therapeutic approaches are urgently needed for remaining patients who do not achieve long-term survival.

While immunotherapy is revolutionizing the treatment landscape for adult patients, its application in pediatric patients remains limited. Currently, immunotherapy is successful in treating children with high-risk neuroblastoma or Non-Hodgkin’s lymphoma using antibodies targeting the ganglioside GD2 or CD20, respectively, and in pediatric B-cell acute lymphoblastic leukemia (B-ALL) using Chimeric antigen receptor-modified T (CAR-T) cells targeting the B-cell biomarker CD19 [2,3,4]. These examples highlight the potential of immunotherapy to improve outcomes for pediatric cancer patients.

In order to accelerate the access of pediatric patients to innovative clinical studies, the European Medicines Agency (EMA) has implemented, since 2007, a pediatric investigation plan (PIP) to support the medicine’s authorization in children. A PIP represents a pediatric investigation concept that ensures the collection of data on the use of medications through clinical studies in children and is intended to guarantee the faster availability of innovative and effective medications for children. To fulfill these requirements, it is of highest interest to know the expression of molecular targets across pediatric tumor entities. Moreover, to predict possible toxicity, it is necessary to know their distribution in normal pediatric tissues as well. While there are several efforts to make genomic and expression data from pediatric tumors available (e.g., the St. Jude Cloud [5]), databases contianing the molecular target expression in normal pediatric tissues are not available. The largest published expression profiling of normal tissues expression, GTEx, lacks publicly available data on the donors’ age, making age-matched studies impossible [6].

Immunotherapy using CAR-T cells targeting GD2 has shown remarkable success in treating pediatric tumors with poor prognosis, such as high-risk neuroblastoma and diffuse midline glioma [7,8]. However, translating this approach to other solid tumors remains difficult, partly due to the scarcity of optimal targets. A major concern is the potential for CAR-T cells to cross-react with healthy tissues, leading to unacceptable toxicity from on-target/off-tumor effects [9]. Consequently, it is essential that the targeted antigens for CAR-T cells are not expressed on normal cells critical for survival. One such antigen candidate is Claudin 6 (CDN6). CLDN6 is an oncofetal tight junction protein expressed in epithelia during organogenesis, but not in healthy adult tissues, except the placenta [10,11,12]. CLDN6 expression is reactivated in various solid adult tumor types, particularly in germ cell tumors (GCTs) and epithelial ovarian cancer, which led to the development of CLDN6-specific CAR-T cells and bispecific T cell-engaging antibodies that are currently being evaluated in basket clinical studies for adult patients [13,14].

Several pediatric tumors emerge from dysregulated embryonic development, which implies the potential overexpression of CLDN6. Indeed, expression of CLDN6 has previously been shown in malignant rhabdoid tumors (MRTs), Wilms tumors (WTs), hepatoblastoma, germinoma and extracranial GCT [15,16,17,18], Thus, CLDN6 could represent an important target for CAR-T cells and monoclonal antibody-based therapies in the pediatric population. Despite these promising data, conflicting results exist on the specificity and sensitivity of CLDN6 immunohistochemistry (IHC), which is commonly used to assess CLDN6 expression. Moreover, a systematic analysis of CLDN6 expression in normal tissues of children at different ages is currently lacking. Compared to adult tissue, pediatric tissue typically contains a high number of self-renewing stem cells and early precursor cells as a physiological resource of growth and regeneration.

Here, we assessed the expression of CLDN6 in normal tissues and tumor samples from children on the RNA and protein levels. IHC was performed with a CLDN6-specific antibody that is currently used in Phase I/II trials to identify patients with CLDN6-positive tumors (NCT05262530; NCT04503278). Our data disclose an age-dependent expression of CLDN6 in normal tissues and an expression of rare but aggressive tumor subtypes, including desmoplastic small round cell tumors (DSRCTs).

## 2. Materials and Methods

### 2.1. Patient Cohort

Surplus tissues from surgeries and from autopsies not needed for histopathological diagnosis were selected by experienced pathologists. In accordance with the ethics committee of Rhineland-Palatinate, the written informed consent of all patients or their custodians was obtained for “scientific use of surplus tissue not needed for histopathological diagnosis”. This study was performed in agreement with the declaration of Helsinki in regard to the use of human material for research. Ethical approval was obtained by the local ethics committee (No. 2021-15871, and No. 2022-16284).

### 2.2. CLDN6 Gene Expression Across Cancer Types

RNA sequencing profiles and corresponding annotation of 1124 samples of 30 blood cancer entities (including 27 leukemia subtypes, two lymphoma subtypes, and Myelodysplastic Syndrome) and 579 samples of 24 extracranial cancer types were obtained from the St. Jude Cloud (https://stjude.cloud, accessed on 30 October 2024). Tumor entities containing less than five samples per category were not extracted. We excluded genes with less than 10 read counts in the sum of all samples. The count matrix was normalized by the median of ratios method using the DESeq2 R package (version 1.34.0) [19]. A pseudocount was incorporated, and, subsequently, the counts underwent logarithmic transformation on a decimal scale. The batch effect caused by different library preparation protocols was removed using the ComBat function from the sva R package (version 3.42.0) [20]. A boxplot of batch-corrected and transformed expression values of CLDN6 was visualized with the ggplot2 R package (version 3.3.6).

### 2.3. CLDN6 Staining

Staining was performed in the central histology lab at BioNTech SE on representative tissue slides of formalin-fixed, paraffin-embedded (FFPE) normal and neoplastic tissues using an in vitro diagnostic kit (CLAUDENTIFY6, BioNTech Diagnostics, Mainz, Germany) which uses the CLDN6-specific antibody clone, 58-4B-2, and a normal control reagent, as previously described [13]. All samples were analyzed by board-certified pathologists regarding tumor content (hematoxylin and eosin) and CLND6 expression in normal and tumor tissues. Only membranous staining was considered. CLDN6 staining was classified as strong intensity (3+), moderate intensity (2+), weak intensity (1+) and absent (0). Score +1 indicated faint/barely perceptible or incomplete membrane staining. Only staining 2+ and 3+ was considered for CLDN6 positivity.

### 2.4. qRT-PCR Expression Analysis

RNA was isolated from fresh-frozen tumor samples using the RNeasy Lipid Tissue Kit (QIAGEN, Hilden, Germany) following the manufacturer’s protocol. The quality of the RNA was determined with a Bioanalyzer Device (Agilent Technologies, Santa Clara, CA, USA), and only samples with adequate RNA integrity number (RIN) values were used for further analysis. Reverse transcription was performed with the PrimeScriptTM RT Reagent Kit with gDNA Eraser (TaKaRa BIO INC, Kusatsu, Japan). Quantitative RT-PCR was performed using the PerfeCTa^®^ SYBR^®^ Green Fast Mix^®^ (Quantabio, Beverly, CA, USA) in a LightCycler 480 instrument (Roche, Basel, Switzerland). After normalization to the housekeeping gene HPRT1, the relative quantification value was expressed as 2−ΔΔCt. The calibrator was calculated as the maximal number of cycles used in the PCR minus the mean of the HPRT1 Ct values. The primers for qRT-PCR analyses were as follows [12]: CLDN6 forward: 5′-ACTCGGCCTAGGAATTTCCCT, CLDN6 reverse 5’-CAGAGGCCATGGCGAGG; HPRT1 forward 5′-TGACACTGGCAAAACAATGCA, reverse 5′-GGTCCTTTTCACCAGCAAGCT. Results were visualized using GraphPad Prism (version 7.02).

## 3. Results

### 3.1. CLDN6 Expression Is Predicted in Solid and Blood Cancer in the Pediatric Population

To identify tumor entities with high expression of CLDN6, we used RNA-sequencing (RNA-seq) data available via the St. Jude Cloud. The expression data of about 500 solid tumor and more than 1000 blood cancer samples across entities were analyzed (Figure 1), excluding intracranial tumors. The highest CLDN6 RNA expression was found in GCTs, particularly of the ovary, DSRCTs and WTs. Interestingly, a high expression of CLDN6 was also seen in some samples of rare blood cancer subtypes, including a core-binding factor AML.

### 3.2. CLDN6 mRNA Is Present in Normal Tissues of Infants and in Pediatric Tumors

Our analysis of RNA-seq data suggested that CLDN6 is an interesting target for several pediatric tumor entities. We validated the expression of CLDN6 in pediatric tumors by qRT-PCR in an independent cohort and compared the expression of CLDN6 in tumors with normal tissues. Only fresh-frozen tissues were used for the analysis, again excluding intracranial normal and tumor tissues. All analyzed tissues are listed in Table 1 (normal tissues) and Table 2 (tumors).

Samples for qRT–PCR–based expression measurements in normal tissues were obtained from newborn children. RNA of acceptable quality (HPRT≤25 and RIN≥ 6) was available for 32 tissues with 45 samples for the age group <1 years. The cohort of extracranial tumor tissues counted initially 160 samples and fairly represented the distribution of pediatric tumor entities in Germany [1] (Appendix A), with the highest percent of samples consisting of bone tumors (22.50%), kidney tumors (18.75%) and peripheral nervous cell tumors (18.13%). Further entities were extracranial germ cell tumors (11.88%), soft tissue sarcomas (15%) and liver tumors (5.63%). The group “carcinomas” included thyroid carcinoma, kidney cell carcinoma and adrenocortical carcinoma (3.75%). The group “other tumor entities” included pleuropulmonary blastoma and nephroblastomatosis (3.70%). Only 0.63% of the cohort consisted of an unspecified group. However, 44 samples had to be excluded because of low RNA quality (HPRT ≥ 25 and RIN ≤ 6). In particular, the quality controls were inadequate for osteosarcoma (50%) and neuroblastoma (33%). Most of these samples (89%) were obtained after a systemic chemotherapy. The distribution of the extracranial solid tumor samples analyzed by qRT–PCR can be found in Figure 2A.

In normal tissues, CLDN6 expression (defined as relative value of ≥10^4^) was detected in samples of newborns (Figure 2B), particularly in the lungs of two donors, in one of two pancreas samples from two donors, one of two colon samples from one donor and one of two skin samples from two donors (Figure 2B). In addition, the renal pelvis sample from a newborn showed the expression of CLDN6 ≥ 10^4^. Renal tissues from newborns that expressed CLDN6 showed a higher expression level in the medulla (two samples from two donors) than in the cortex.

In tumor samples, the highest expression of CLDN6 was detected in extracranial GCT, the DSRCT sample available in our cohort and nephroblastoma (Figure 2B). Extracranial GCT consisted of 13 samples from 11 different patients in the age category of 0 up to 9 years. Yolk sac and mixed-type groups showed the highest expression of CLDN6 (Figure 2C). Within teratoma, only immature but not mature teratoma expressed CLDN6. In yolk sac tumors, the expression of CLDN6 appeared to decrease with the patient’s age. However, due to the limited number of samples, it is not possible to draw definitive conclusions regarding a potential correlation between CLDN6 expression and age (Appendix A). The DSRCT sample was retrieved from a 16-year-old patient and showed the second highest expression above a value of >10^5^ (Figure 2B, in the category “other tumor entities”)

Nephroblastoma consisted of 20 samples from 15 patients. Eight samples from six patients (40%) showed a positive expression of CLDN6 with a value of about ≥10^4^ (Figure 2D). Histologically, these samples belonged to epithelial, stroma-type, mixed-type, regressive, focal anaplasia and diffuse anaplasia WTs (Figure 2B,D). Two samples (10%) of different patients showed a positive expression of CLDN6 with a value of about ≥10^5^. Histologically, they belonged to the mixed and diffuse anaplasia type. Five samples from two patients were isolated from metastasis (Figure 2B). Among them, three samples from one patient were isolated from the liver and lungs and were negative. They belonged to the mixed type. The corresponding primary tumor sample was also negative. Two samples from another patient isolated from the chest wall and lymph node were positive and belonged to diffuse anaplasia. The corresponding primary tumor sample was not analyzed. In one case, a WT and nephroblastomatosis were analyzed in the same patient, with the highest expression being found in the WT component (Figure 2B).

MRTs consisted of five samples from two patients. One sample showed high expression of CLDN6 with a value of about ≥10^4^. This sample corresponded to a lymph node metastasis after systemic chemotherapy. Negative samples were isolated from the primary tumor located in the cervical area and in the liver (Figure 2B).

Ewing sarcoma (nine samples from eight patient), osteosarcoma (eleven samples from eight patients), rhabdomyosarcoma (eight samples from six patients) and neuroblastoma (16 samples from 13 patients) showed a low expression of CLDN6 below a value of 10^4^.

The expression of CLDN6 in the category “other tumor entities” (see Table 2 for details) was <10^4^, with the exception of the mentioned DSRCT sample.

Taken together, these results suggest that CLDN6 RNA is still present in several normal tissues in newborns and confirms the expression in DSRCTs, the subtypes of germ cells and WTs.

### 3.3. Organ–Specific Presence of CLDN6-Positive Epithelial Precursors

The CLDN6 expression we detected in newborn tissues is likely a reflection of its functional role during early developmental stages. CLDN6 is indeed involved in the early epithelialization of the mouse embryo, and expression in human fetal tissues has previously been described in the epithelium of lungs and the kidney [10,15]. In this study, we expanded the analysis of CLDN6 to a broader range of fetal tissues and performed the staining with the antibody currently used to enroll patients in clinical studies targeting CLDN6 with CAR–T cells and bispecific antibodies [13,14]. Strong (score 3+) and continuous (100%) expression was detected in the epithelium of skin, the proximal tubuli of the kidney, the lungs, the intestinal tract and pancreas (Figure 3 and Figure 4 and Table 3). Strong expression in at least 40% of the cells was detected in the mesothelial cells and yolk sac. Strong and continuous homogeneous expression was found also in the placental epithelium. The embryonal epithelium of the thyroid gland, adrenal glands, and gonads were completely CLDN6-negative (Appendix A). Taken together, these results indicate the presence of CLDN6-positive precursors in the primitive squamous and glandular epithelium of only a few embryonic organs, suggesting a key role for CLDN6 in the process of organ-specific epithelial differentiation during the early development of these organs.

### 3.4. CLDN6 Expression Is Lost After Birth

In the embryo, CLDN6 is expressed in various tissues. Therefore, before including children in clinical studies with therapies targeting CLDN6, it is crucial to determine when CLDN6 expression diminishes after birth. We analyzed CLDN6 expression by IHC in four age groups and in different tissues, including all tissues which were positive for CLDN6 in the embryo (Table 4, Figure 4 and Appendix A). We detected a small population of CLDN6-positive epithelial progenitors present only in the first few weeks of the child’s life. In children above 1 year of age, we observed only isolated, difficult-to-detect clusters of CLDN6-positive progenitors. The transient expression of CLDN6 during postnatal and prenatal development is exemplified for the kidney (Table 4 and Figure 4). While undifferentiated cells (blastema) were negative, epithelial cells of tubuli strongly expressed CLDN6, but the expression decreased from more than 80% positive cells in the first 10 weeks of pregnancy to approximately 5% in the 37^th^ pregnancy week. Moreover, one part of the immature tubular epithelium was CLDN6-negative, while the other was positive, which likely reflects the asynchronous differentiation of the renal epithelium and features of kidney development. Taken together, these data indicate that, with the exception of the first week of life, CLDN6 expression is lost after birth.

**Figure 4 cancers-17-00920-f004:**
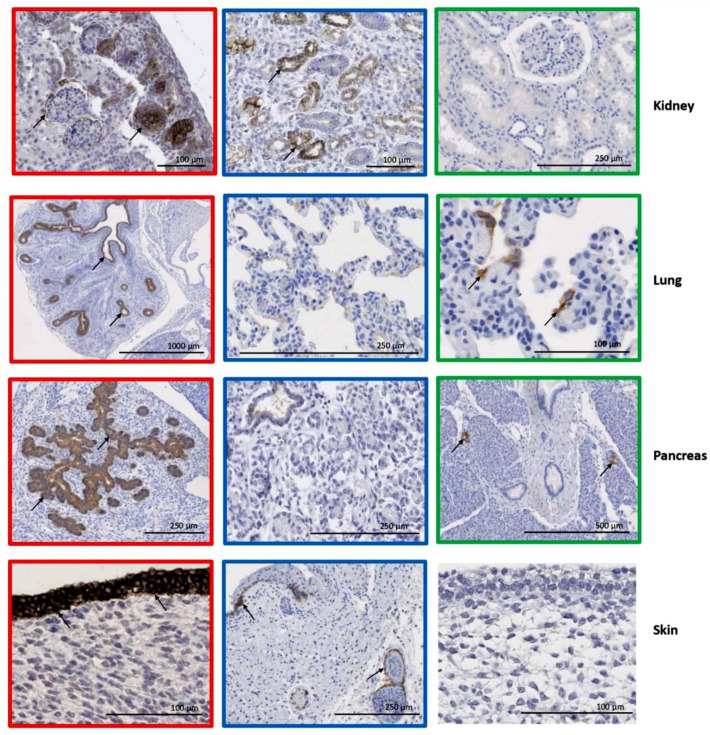
CLDN6 expression is lost after birth. IHC was performed with the indicated organs and in different age groups, including embryo. The picture frame in red indicates tissues isolated from an embryo at the fifth week of pregnancy, while blue indicates a newborn (1 week old) and green indicates 13- to 18-years-old donors. Expression of CLDN6 in sparse small clusters can be observed in pancreas, skin, kidney and lung after birth. Negative control (embryonic skin, right without frame). The arrows indicate the CLDN6 positive epithelial cells.

**Table 4 cancers-17-00920-t004:** Transient expression of CLDN6 in kidneys during normal development. The % of CLDN6-positive cells is reported as mean and standard deviation. Abbreviations: Mo/y.—donor age (months and years, respectively). Score interpretation: green: negative (0, light green and 1+ darker green); yellow: 2+; orange: 3+.

Localization/Differentiation	Prenatal Development (n = 14)			Postnatal Development (n = 29)				
	I Trimester	II Trimester	III Trimester	0–1 Mo	1–12 Mo	1–5 y	6–12 y	13–18 y
**Cortex**								
**Undifferentiated cells (blastema)**	1.1 ± 1.0	0	0	0	-	-	-	-
**Epithelial cells**								
Tubuli	81.3 ± 7.0	34.3 ± 21.5	10.7 ± 8.2	5.0 ± 0.5	7.3 ± 4.5	5.4 ± 2.7	0.2 ± 0.4	0
Bowman’s capsule	14.0 ± 8.5	9.3 ± 13.9	0	1.0 ± 0.3	0.3 ± 0.8	0	0	0
**Stromal cells**								
Glomerular capillaries	3.0 ± 0.6	0	0	0	0	0	0	0
Connective tissue cells	0	0	0	0	0	0	0	0
**Medulla**								
**Epithelial cells**								
Collecting Tubuli	92.5 ± 3.5	34.3 ± 27.8	8.3 ± 2.9	5.0 ± 0.5	2.7 ± 3.3	0.7 ± 1.7	0	0
**Stromal cells**								
Connective tissue cells	0	0	0	0	0	0	0	0

### 3.5. CLDN6 Is Expressed at the Protein Level in GCTs, DSRCTs and WTs

To confirm that CLDN6 mRNA expression results in protein expression in tumor cells, CLDN6 expression was analyzed by IHC. Tumor samples analyzed by IHC are listed in Table 2. Strong and homogeneous expression of CLDN6 was detected in the DSRCT sample (Figure 5A) and in the GCT sample (Figure 5B). CLDN6 expression was also found in nephroblastoma but was rather heterogeneous (Figure 5C,E,F).

In extracranial GCTs (six samples from five different patients), five samples (83%) showed positivity membrane cells from 80 to 95%. In line with mRNA expression data, one mature teratoma sample from a 14-year-old female patient was negative for CLDN6 membrane staining (Figure 5 and Appendix A).

In nephroblastoma (13 samples from 11 different patients), positivity was detected in 27% of the patients (Figure 5 and Appendix A). Two primary tumor samples from two different patients showed membrane staining in 25–50% of the cells (intermediate risk). Six samples isolated from primary tumors from six patients showed an insignificant membrane staining (score from 0+ to 1.5+) in 0–5% of the tumor cells. One metastasis sample and one blastemic-type sample showed an unspecific nuclear staining. Two relapse samples from one patient showed an insignificant staining in 0.5–1% of the cells. In samples from high-risk patients (n = 3), one sample isolated from a lymphnode metastasis showed a high CLDN6 expression, but in only 5% of the tumor cells. Another sample was CLDN6 negative, and another sample showed CLDN6 positivity only in a part of the tumor epithelium, while the blastemal component was completely CLDN6-negative (Figure 5C,E,F). Intermediate-type samples (n = 9) from seven patients showed a positive CLDN6 expression in two patients. In one sample, the risk type was not defined. One nephroblastomatosis sample showed strong CLDN6 expression in 50% of the immature epithelial (Figure 5D).

CLDN6 membranous staining was absent in MRTs (five samples from four different patients), osteosarcoma (50 samples from 12 patients), rhabdomyosarcoma (64 samples from 21 different patients), neuroblastoma (110 samples from 24 different patients) and in two mesoblastic nephroma samples from two patients (Figure 5 and Appendix A).

Matched samples analyzed by qRT–PCR and immunohistochemistry were available for five extracranial germ cell tumors, eight nephroblastoma, two malignant rhabdoid tumors, three osteosarcoma, three rhabdomyosarcoma, eight neuroblastoma, two mesoblastic nephroma and one nephroblastomatosis (Appendix A). All samples with a high number of CLDN6-positive tumor cells (five extracranial germ cell tumors, two nephroblastoma and the one DSRCT) expressed also CLDN6 mRNA at >10^4^. In contrast, no membranous staining was observed if CLDN6 mRNA was not detectable (25 samples). Some samples (n = 4) showed a nuclear or cytoplasmatic staining, but not CLDN6 mRNA expression, suggesting that any staining other than membranous is likely an off-target artifact. In one primary WT sample and one nephroblastomatosis, mRNA expression was not able to predict CLDN6 membranous protein expression. The WT sample was at the lower limit of the quality control (RIN 6.8) and was retrieved after the patient received systemic chemotherapy. In another primary WT sample, mRNA expression was able to predict CLDN6 protein expression even though only 25% of the cells were positive in IHC. One metastasis sample with high expression of CLDN6, but in only 5% of the cells, was also identified as CLDN6-positive by qRT–PCR.

Taking both qRT–PCR and IHC results into account, CLDN6 expression was detected in 50% of 12 GCT patients, 25% of four MRT patients and 53% of 19 WT patients (Appendix A).

In conclusion, CLDN6 membranous expression is found in GCTs, WTs and DSCRTs. However, homogeneous expression was found only in GCTs and DSCRTs.

## 4. Discussion

CAR-T cells eradicate tumors by targeting specific surface markers, harnessing the patient’s own immune system to recognize and destroy cancer cells. This technology has shown tremendous potential, particularly in hematologic malignancies amongst the adult and pediatric populations. While this approach is still limited for use in solid tumors for multiple reasons [3,21], recently, the first promising results were obtained for CAR-T cells targeting GD2-expressing tumors like neuroblastoma and high-grade glioma in children [7,8]. Due to the efficacy of engineered T cells, life-threatening on-target toxicity can occur if the antigen is expressed on normal tissues [9], as shown, for example, in the tight junction proteins CLDN18.2 [22,23] and HER2 [24].

CLDN6 is expressed in tumors and embryonic tissues, but not in the normal tissues of adult donors. Promoter methylation has been suggested to regulate CLDN6 expression [25]. The function of CLDN6 also strengthens its relevance as a target. CLDN6 is prominently expressed in early embryonic development and contributes to the maintenance of the pluripotent state in stem cells [26]. During differentiation, CLDN6 expression is downregulated, correlating with the transition from a pluripotent state to more specialized cell types. Therefore, targeting CLDN6 will presumably contribute to eliminating less differentiated and more aggressive tumor cells [27]. Moreover, CLDN6 supports cell migration and proliferation and can enhance chemoresistance, and its expression is associated with the patient prognosis of a variety of tumors [28,29,30]. Targeting antigens with an important role in tumor’s viability limits the potential for cancer cells to develop escape variants. However, a dual role of CLDN6 in tumor progression has also been discussed, and functional studies addressing the role of CLDN6 in pediatric tumors should be performed in the future [25]. Interim analyses from an ongoing phase I/II trial (NCT04503278) showed significant clinical activity in CAR-T cells targeting CLDN6 when combined with CLDN6-encoding CAR-T cells amplifying RNA vaccine in adult GCT and ovarian carcinoma patients without any evidence of meaningful on-target/off-tumor toxicity [13].

In line with the previous literature, our data show an organ-specific presence of CLDN6-positive epithelial precursors in the early development phase and suggest that CLDN6 is a marker of intermediate, immature differentiation of the epithelium of some organs. Indeed, we demonstrated the absence of CLDN6 in undifferentiated blastema cells as well as in terminally differentiated renal epithelium in different pediatric age groups. CLDN6 expression showed intra-organ epithelial-marked heterogeneity. As an example, one part of the immature tubular epithelium was CLDN6-negative, while the other was positive, which likely reflects asynchronous differentiation of the renal epithelium and features of kidney development. We observed a gradual decline in the relative number of CLDN6–positive epithelial progenitors during pregnancy and a significant decline after birth. We detected a small population of CLDN6-positive epithelial progenitors present only in the first few weeks of the child’s life. In older-age groups of children, we observed only isolated, difficult-to-detect clusters of CLDN6-positive progenitors. It is possible that these cells are resident progenitor cells that are activated under regenerative conditions.

The highest expression in terms of percentage of positive tumor cells and staining intensity was observed in DSRCTs. Only one sample from a 16-year-old patient was available for IHC analysis in our cohort, however high and homogeneous CLDN6 expression has been recently described in another DSRCT patient [13]. Moreover, the RNA-seq data analyzed in our work also predict high levels of CLDN6 in this entity. DSRCT is an extremely rare, aggressive sarcoma diagnosed predominantly in male adolescents and young adults. However, patients younger than 10 years of age can also be affected. Despite aggressive multimodal treatment, DSRCT patients have an extremely poor prognosis, with a 5-year overall survival of about 20% [31]. The majority never reaches complete remission (CR). However, even after CR, approximately 75% of the patients relapse within 3 years after diagnosis [32]. Histologically, DSRCTs appear as nests of “small round blue cells” surrounded by a dense desmoplastic stroma. Immunohistochemical analysis suggests multi-lineage differentiation that includes co-expression of epithelial, mesenchymal, and neuronal markers and indicates that the cell of origin of DSRCTs is a progenitor cell with potential for multiphenotypic differentiation [33]. The EWS–WT1 transcription factor is the driver of tumorigenesis in DSRCTs and it acts by upregulating the expression of several growth factors [34].

Strong and homogeneous staining was observed also in GCTs, in line with previously published data [13]. In contrast to DSRCTs, malignant germ cell tumors show a significantly higher prevalence in the pediatric population and are mostly curable with conventional multimodal treatment even in advanced metastatic disease and after relapse. However, certain subgroups like primary mediastinal GCTs and advanced stage GCTs in children above 10 years of age are associated with dismal outcome and thus require novel treatment strategies [35,36].

The third entities showing CLDN6 expression in our study is WTs, which result from aberrant differentiation of nephric stem cells. WTs predominantly affect children under five years and consist of three elements, represented in varying proportions: poorly differentiated cellular blastema, stroma, and epithelium with tubular formation. WTs with a blastemal–predominant pattern are challenging to distinguish from DSRCTs [37]. WTs and DSRCTs share the involvement of the Wilms tumor gene, WT1, in their pathogenesis. WT1 is a transcription factor that may play a role in mediating the shift from a mesenchymal to an epithelial phenotype [38]. DSRCT is characterized by the fusion of the EWSR1 amino-terminus to the WT1 carboxy-terminus, leading to an aberrant gene expression. Concerning WTs, about 10–15% of cases arise through constitutional or somatic mutations in the WT1 suppressor gene. Inactivation/mutation of the WT1 gene is involved in the development of a subtype of WTs with a stromal-predominant or stromal-type histology and often aberrant mesenchymal cell types, in line with evidence that loss of WT1 function may lead to the activation of ectopic myogenesis [39]. Therefore, we hypothesize a negative correlation between WT1 loss of function mutations or deletions and CLDN6 expression in WTs, which should be tested in future analyses. It is important to note that absent expression of CLDN6 was observed in undifferentiated regions of the blastema of normal embryonic organs. Consistent with this result, we did not detect CLDN6 expression at blastema sites in high-risk nephroblastoma tumor tissues. Our cohort mainly included primary tumor samples; samples at the stage of relapse and metastases should be included in future analysis.

In contrast to Sullivan et al. [15], we did not confirm a high frequency of expression in MRTs by IHC, although whole-mount sections were used. Only one sample obtained from a cervically localized lymph node metastasis expressed high levels of *CLDN6* RNA. Although the number of MRT samples analyzed in our work is too low to allow for final conclusions about CLDN6 staining in this entity, discrepancy on the frequency of positive samples within intracranial rhabdoid tumors (AT/RTs) has been reported as varying from 100% [16] to 29% [17] and 39% [15]. Possible explanations include the cross-reactivity of the CLDN6 antibody with other claudin proteins due to the high homology between different claudins, particularly when polyclonal antibodies are used, as in the case of the three studies mentioned before. In our case, a monoclonal antibody was used (clone 58–4B–2) that is currently used to identify patients with CLDN6-positive tumors in Phase I/II trials with T cell–engaging bispecific antibody and CAR–T cells (NCT05262530; NCT04503278).

Based on transcriptomic data analyzed in this work, CLDN6 expression is expected in several samples of rare subtypes of acute myeloid leukemia (AML), namely acute megakaryoblastic leukemia and core-binding factor AML. Both entities are heterogeneous, and, particularly in the adult population, a substantial proportion of patients relapse and eventually die, emphasizing the need for further therapy improvements [40,41]. Validation at the protein level will be necessary to clarify the relevance of CLDN6 in the pathology and therapy of these rare diseases.

In this study, only extracranial normal and tumor tissues were analyzed. Because CAR–T cells can cross the blood–brain barrier, it is imperative to extend the analysis of CLDN6 expression to the normal brain tissues of children at different ages in future studies. This is particularly important because the highly aggressive intracranial pediatric tumor AT/RT expresses CLDN6 and this entity occurs most commonly in infants and toddlers [15]. Other intracranial tumor entities that may express CLDN6 are germinoma and primitive neuroectodermal tumor [15,16]. Due to the limited number of samples in our cohort, we were unable to analyze the correlation between CLDN6 expression and tumor stage or histological subtypes. However, a previous study has demonstrated significant correlations between CLDN6 expression, molecular subtypes, and clinical characteristics in various adult cancers, particularly in endometrial carcinoma, using transcriptomic data [30].

Pre-screening with qRT–PCR seems to be a valuable method to identify CLDN6-positive samples in cases where highly specific CLDN6 antibodies are not available. Our results show that qRT–PCR data correlate well with IHC data if the tumor is CLDN6-negative or CLDN6 expression is homogeneous. Samples with nuclear or cytoplasmic CLDN6 expression were negative in qRT–PCR, suggesting unspecific staining in these cases, even if nuclear expression of claudins has been reported [42]. However, in several cases, the qRT–PCR analysis was impaired due to poor quality of the RNA, particularly if the samples were collected after chemotherapy. Moreover, if the expression of CLDN6 is heterogeneous, qRT–PCR alone will not be sufficient to identify samples with a therapy-relevant CLDN6 expression. Flow cytometry and single-cell transcriptomics could be employed to investigate the heterogeneity of CLDN6 expression, enabling quantification and analysis of its co-expression with other markers [43]. These approaches could provide new insights into the variations in CLDN6 levels across different cell populations.

In terms of CLDN6-targeting therapies, CAR-T cells, T cell–engaging bispecific antibodies and Antibody Drug Conjugates (ADCs) are in clinical trials, and the first results with CAR–T cells have been published [13,14,44]. Objective response with CAR–T cells occurred only in patients with either epithelial ovarian cancer or GCTs [13]. In that small cohort of patients (n = 21), all responding patients had tumors with >80% of tumor cells expressing 2+/3+ CLDN6 at prescreening, suggesting that the CLDN6 expression level may be predictive of the outcome. In pediatric tumors, this condition applies only to DSCRTs and GCTs, while, in WTs, the expression is too heterogeneous, reflecting the histological heterogeneity of WTs, which is reminiscent of embryonal kidney development. Particularly high-risk WTs are characterized by blastemal tumor cells, which are CLDN6-negative. High-risk WTs with diffuse anaplasia are potentially CLDN6-positive, and more samples from patients in relapse under chemotherapy should be analyzed in the future. However, CLDN6 expression alone cannot explain the success of the CAR-T cell treatment. In the published data, four GCT patients had progressive disease (PD) despite high CLDN6 expression. Moreover, one DSRCT patient with high expression of CLDN6 showed no objective response to the treatment [13]. Other factors such as the persistence of CAR-T cells and the dosage may influence the success of the therapy. Moreover, loss of CLDN6 expression under therapy pressure has been described in preclinical studies, and future work will be required to understand and modulate CLDN6 expression in different tumor types [45].

In terms of expected toxicity, the immune system of pediatric patients may respond more vigorously or differently than that of adults to foreign proteins or engineered immune therapies, increasing the likelihood of immune-mediated adverse events. These events can include cytokine release syndrome (CRS) and immune effector cell-associated neurotoxicity syndrome (ICANS). Notably, CRS was observed in 46% of adult patients treated with CAR-T cells targeting CLDN6 [13]. Given the potential for such reactions, it is essential to implement supportive care measures to manage and mitigate these adverse effects effectively. Finally, the high degree of structural and sequence similarity within members of the CLDN family raises the question of possible off-target toxicities that occur when the transduced T–cell population unexpectedly attacks an antigen other than the one that was intended. Notably, the extracellular regions of CLDN6 and CLDN9 vary by just three amino acids, and several antibodies failed to show selective binding to CLDN6 [46]. The anti-CLDN6 CAR-T cells identify the antigen using a single-chain variable fragment derived from the monoclonal antibody IMAB206–C46S. Although CLDN9 is expressed in healthy tissues, such as the pituitary gland, no off-target toxicity has been reported, suggesting the antibody’s specificity [13,47].

## 5. Conclusions

In conclusion, this work supports the importance of CLDN6 in the normal epithelialization process, explaining its maintenance in a subset of pediatric tumors that develop from embryonal progenitors. These tumors can hijack developmental programs, causing a block in differentiation and, as a consequence, unrestricted proliferation. Because the expression of CLDN6 is lost very quickly after birth, children suffering from GCTs and DSRCTs might benefit from anti-CLDN6 based therapies such as CAR-T cells, which are currently tested in the adult population.

## Figures and Tables

**Figure 1 cancers-17-00920-f001:**
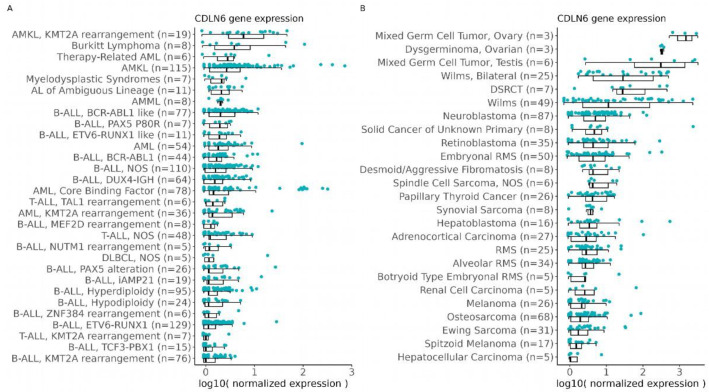
CLDN6 expression is predicted in solid and blood cancer in the pediatric population. Batch-corrected and transformed gene expression values of CLDN6 across blood cancer (**A**) and extracranial cancer types (**B**). The lower half represents a boxplot per tumor type, where the box range from the first to third quantiles, divided by a line indicating the median, and whiskers demonstrate the largest and lowest values no further than 1.5 × IQR from the hinge. The upper half shows the distribution of the samples, where each dot represents a sample. B-ALL = B-cell Acute Lymphoblastic Leukemia, T-ALL = T cell Acute Lymphoblastic Leukemia, AML = Acute Myeloid Leukemia, AMKL = Acute Megakaryoblastic Leukemia, AMML = Acute Myelomonocytic Leukemia, AL = Acute Leukemias, DLBCL = Diffuse Large B-cell Lymphoma, RMS = Rhabdomyosarcoma, DSRCT = Desmoplastic Small Round Cell Tumor.

**Figure 2 cancers-17-00920-f002:**
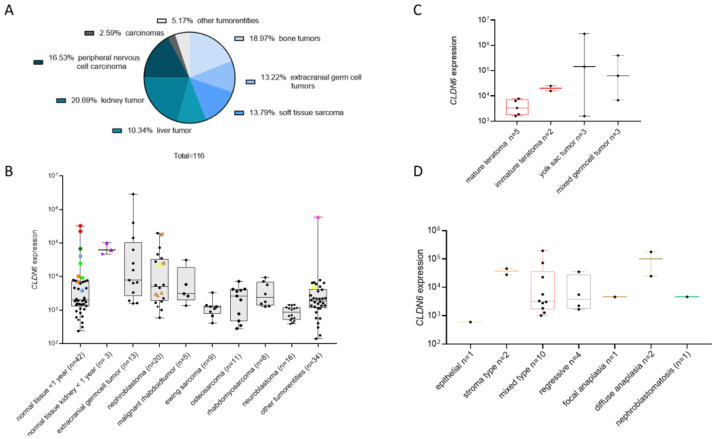
CLDN6 mRNA is present in the normal tissues of infants and in pediatric tumors. (**A**) Distribution of the solid tumor samples analyzed by qRT–PCR. (**B**) Expression of CLDN6 analyzed by qRT–PCR in normal and tumor tissues. Relative expression is shown as a box and whisker plot; the box range is from minimum to maximum value. Each dot represents a sample. In the normal tissues boxplot, red dots indicate lung, purple dots indicate kidney, bright green dots indicate pancreas, blue dots indicate skin, dark green dots indicate renal pelvis and orange dots indicate colon samples. In the normal kidney boxplot, a purple triangle indicates the medulla and the star cortex samples. In the boxplot for “nephroblastoma,” the brown dots and triangles represent metastases from two different patients, while the yellow triangle indicates the nephroblastoma component of a tumor that also presented with nephroblastomatosis (depicted as a yellow dot in the “other tumor entities” boxplot). Further, in the boxplot “other tumor entities”, a pink dot represents a DSRCT sample. (**C**) Relative expression of CLDN6 in subtypes of GCT. (**D**) Relative expression of CLDN6 in subtypes of nephroblastoma.

**Figure 3 cancers-17-00920-f003:**
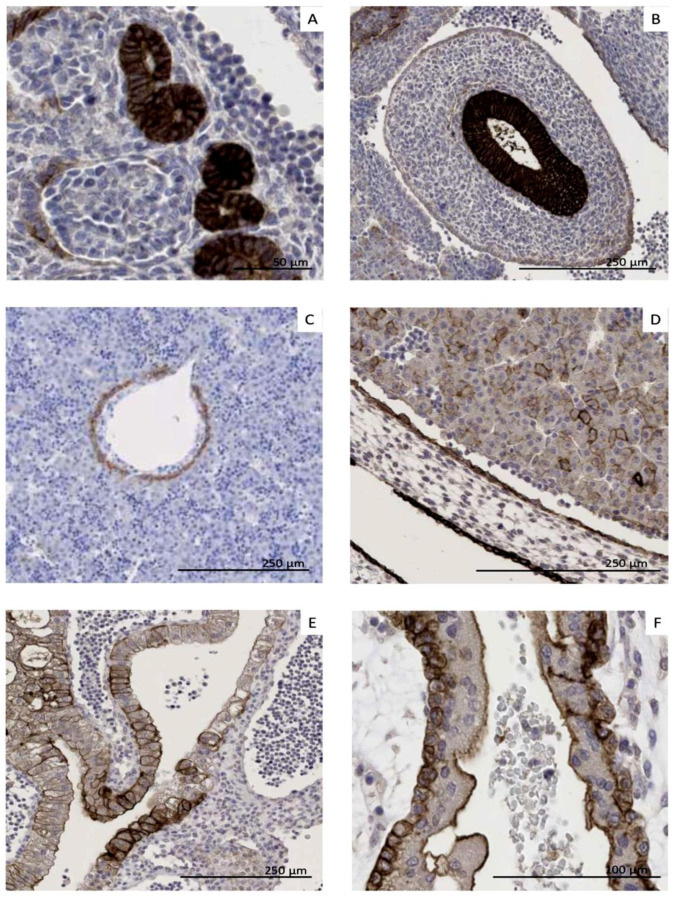
Organ–specific presence of CLDN6–positive epithelial precursors (5th week of pregnancy). Tissues were stained with an anti–CLDN6 specific antibody. The evaluation of the staining can be found in Table 4. Embryonic epithelium of (**A**) kidney (score 2+/3+), (**B**) Intestine (score 3+), (**C**) liver ducts (score 2+/3+), (**D**) peritoneum (score 2+). (**E**) Yolk sac (score 2+/3+). (**F**) Chorionic epithelium of the placenta (score 2+/3+).

**Figure 5 cancers-17-00920-f005:**
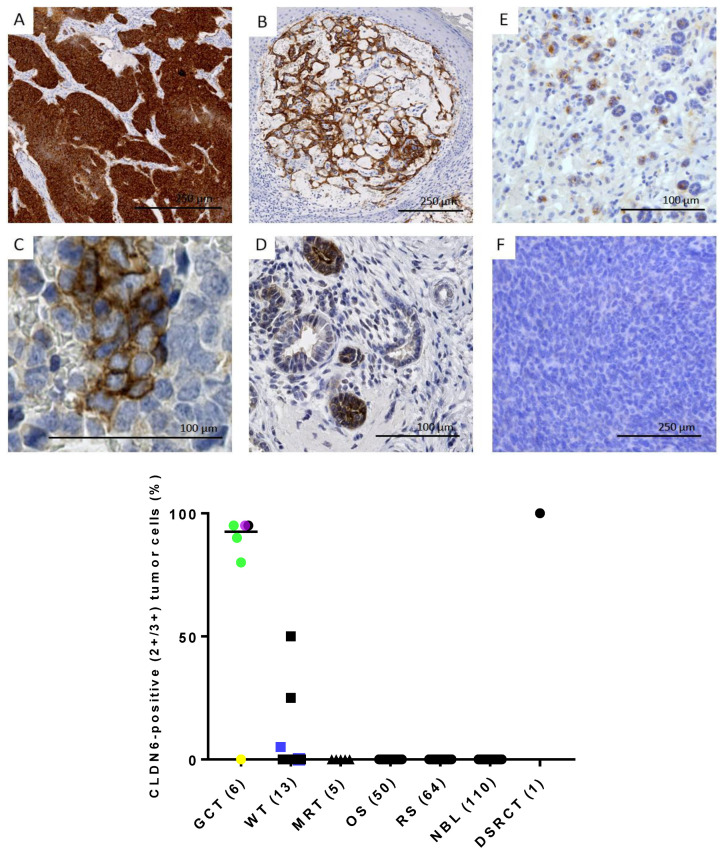
CLDN6 was analyzed by immunohistochemistry in DSRCT ((**A**), 100% of tumor cells are positive, with score 3+); Yolk sac tumor ((**B**), 95% of the tumor cells are positive, with score 2+/3+) and high-risk WTs after therapy (**C**,**E**,**F**). Blastema (**F**), tumor stroma and part of the tumor epithelium (**E**) were CLDN6-negative, individual tumor cells in the metastasis were positive (**C**), 5%. The persistent immature nephroblastomatosis foci were partially positive (**D**). In the graph, the intensity and heterogeneity of CLDN6 expression across pediatric tumor entities is reported. The y-axis indicates the percentage of positive tumor cells (with a score of 2+/3+). Each dot represents a sample. The median value is indicated. Within GCT, the yellow dot indicates a mature teratoma, the green dots yolk sac tumors, the pink dot a dysgerminoma and the black dot a mixed subtype. Within “WT”, the blue dots indicate high-risk patients.

**Table 1 cancers-17-00920-t001:** Normal tissues used for the analysis of CLDN6 expression by qRT-PCR and IHC. The number of analyzed tissues is indicated. If not indicated diversely (number in bracket), tissues from one donor were analyzed. y = years.

Nr	Tissues	qRT-PCR	Immunohistochemistry			
		<1 y	<1 y	1–5 y	6–12 y	13–18 y
1	Adipose tissue skin		4	2	2	2
2	Adipose tissue visceral	1	1	2	2	2
3	Appendix	1	1	1	1	1
4	Connective tissue visceral			2	2	2
5	Blood vessels	1	1	2 (n = 2)	2	2
6	Adrenal gland	1	2 (n = 2)		2	2
7	Bladder					
8	Bone Marrow	2 (n = 2)	2	2	2	2
9	Bone growth plate	2 (n = 2)	2	2	2	
10	Colon	2	2	2	2	2
11	Duodenum		2	1	1	1
12	Esophagus	2 (n = 2)	2 (n = 2)	1	1	1
13	Epididymis	1	2	2	2	2
14	Fallopian tube		2 (n = 2)	1	1	1
15	Gall Bladder	1	2	1	1	1
16	Gastrointestinal tract					
17	Heart	3 (n = 2)	2	4	4	4
18	Ileum	1	1	1	1	1
19	Kidney Cortex	1	2	2	2	2
20	Kidney Medulla	2 (n = 2)	2	2	2	2
21	Kidney					
22	Liver		2	2	2	2
23	Lung	2 (n = 2)	2	2	2	2
24	Lymph node		2	2	2	2
25	Nerve	1	2 (n = 2)	2 (n = 2)	2	2
26	Ovary		2 (n = 2)		2	2
27	Pancreas	2 (n = 2)	2	2		2
28	Placenta 8w		2 (n = 2)			
29	Placenta 40W		2			
30	Prostate		2			1
31	Rectum		2 (n = 2)			
32	Renal pelvis	1				
33	Retina		1			
34	Salivary Gland		2	1	1	1
35	Skin	2 (n = 2)	2 (n = 2)	2	2	2
36	Small intestine			2	2	2
37	Spleen	2 (n = 2)	2 (n = 2)	1	1	1
38	Stomach	1	2 (n = 2)	1	1	1
39	Smooth muscle		4 (n = 3)	1	1	1
40	Skeleat muscle	5 (n = 5)	2 (n = 2)	2 (n = 2)	2 (n = 2)	2
41	Striated muscle					
42	Testis	3 (n = 3)	2	2	2	2
43	Thymus	2 (n = 2)	2	2	2	2
44	Thyroid		2 (n = 2)	2	2	2
45	Tibia					
46	Tonsil		2 (n = 2)	2	2	2
47	Trachea	1	1	1		1
48	Ureter	2 (n = 2)	2	2	2 (n = 2)	2
49	Uterus		2 (n = 2)	2 (n = 2)	2	2
50	Vagina		2	2 (n = 2)	2	2
51	Bronchus		1	1	2	1
52	Sigmoid colon		2 (n = 2)			

**Table 2 cancers-17-00920-t002:** Tumor tissues used for the analysis of CLDN6 expression. The number of tissues and donors (in brackets) is indicated. In total, 116 tissues from 88 patients and 254 tissues from 79 patients were analyzed by qRT–PCR and IHC, respectively.

	qRT–PCR		IHC		
Entity	Tissues	Age	Tissues	Age	Comment
extracranial germ cells tumors	13 (n = 11)	8 (0–9)	7 (n = 5)	4 (1–14)	One sample from relapse
nephroblastoma	20 (n = 15)	3 (0–12)	13(n = 10)	6 (1–10)	Three samples from metastasis, two samples from relapse
malignant rhabdoid tumor	5 (n = 2)	2	5 (n = 4)	2 (1–16)	
Ewing sarcoma	9 (n = 8)	14 (4–16)	0		
osteosarcoma	11 (n = 8)	16 (10–20)	50 (n = 12)	16 (11–21)	Eleven samples from lung metastasis, 12 samples from relapse (six soft tissue periprosthetic, four lung, two thoracic wall and lung)
rhabdomyosarcoma	8 (n = 6)	4.5 (1–13)	64 (n = 21)	5 (1–15)	one sample from lymphnode metastasis
neuroblastoma	16 (n = 13)	2.5 (0–10)	110 (n = 24)	1 (0–12)	Twenty-nine samples from metastasis, 18 samples from relapse
Other tumorentities:	34 (n = 25)				
paraganglioma	2 (n = 2)	11.5 (9–14)	0		
liver sarcoma	2 (n = 1)	15	0		
adamantinoma	2 (n = 1)	4	0		
hepatoblastoma	6 (n = 4)	5.5	0		
adrenocortical tumor	2 (n = 2)	7 (3–11)	0		
pleuroplumonary blastoma	6 (n = 1)	5 (5–6)	0		
neurinoma	1	15	0		
fibromyxoid sarcoma	1	14	0		
undifferentiated sarcoma	2 (n = 2)	15 (1–16)	0		
fibrosarcoma	1	1 (4)	0		
HCC	1	17	0		
kidney cell carcinoma	1	11	0		
mesoblastic nephroma	3 (n = 3)	1 (1–10)	2 (n = 2)	1	
GIST	1	16	0		
neuroendocrine tumor	1	11	0		
DSRCT	1	16	1	16	
nephroblastomatosis	1	3	1	3	

**Table 3 cancers-17-00920-t003:** Organ-specific expression of CLDN6 in normal pediatric tissues. Tissues isolated from four age groups and embryo (<10 week of pregnancy) were isolated. The % of CLDN6-positive cells is reported. Score interpretation: white: not analyzed; green: negative (0, light green and 1+ darker green); yellow: 2+; orange: 3+. n indicates the number of samples analyzed across the five groups.

Localization (n)	<10 w	0<1 y	1–5 y	6–12 y	13–18 y	Localization	<10 w	0<1 y	1–5 y	6–12 y	13–18 y
**Epithelium**						**Lymphoid tissue**					
Esophagus (5)						Spleen (5)					
Intestinum (20)	**100%**					Tonsils (8)					
Liver ducts (9)	**90%**	**1–4%**			**0–1%**	Thymus (8)					
Liver hepatocytes (9)	**10%**					Lymph nodes (8)					
Gall bladder (5)											
Pancreas ducts (7)	**100%**	**2%**			**0–2%**	**Other tissue**					
Pancreas acini (6)		**2–5%**			**0–1%**	Heart (8)					
Pancreas islets (6)						Adipose tissue (10)					
Lung (9)	**100%**	**0–2%**			**0–4%**	Connective tissue (8)					
Bronchus (5)						Smooth muscle (9)					
Kidney (41)	**88%**	**0–5%**				Skeletal muscle (9)					
Prostate (3)						Peripheral nerves (7)					
Ureter (8)						Endothelium		**1–11%**			
Salivary Gland (5)						Blood cells					
Skin (9)	**100%**	**5–10%**			**0–2%**						
Mesothelial cells (1)	**50%**										
Thyroid gland (8)			**0–1%**								
Adrenal gland (8)											
Ovary (6)											
Testis (8)		**1%**	**1–3%**								
Epididymis (8)											
Uterus (8)											
Fallopian tube (5)											
Vagina (8)											
Placenta (1)	**100%**										
Yolk sac (1)	**50%**										

The number of donors and samples analyzed in each age group is shown in Table 1. Abbreviations: %–min–max range of relative number of positive cells, w–week of pregnancy, y–donor age.

## Data Availability

The following St. Jude Cloud data sets were used: Pediatric Cancer Genome Project (PCGP) and Childhood Solid Tumor Network (CSTN): This study used data generated by the St. Jude Children’s Research Hospital—Washington University Pediatric Cancer Genome Project and Childhood Solid Tumor Network [48]. Genomes for Kids (G4K): This study used data generated by the St. Jude Children’s Research Hospital Genomes for Kids Study [49]. Real–Time Clinical Genomics (RTCG): This study makes use of data generated by St. Jude Children’s Research Hospital [50]. Pan–Acute Lymphoblastic Leukemia (PanALL): This study makes use of data generated by the Pan–Acute Lymphoblastic Leukemia Data Set of St. Jude Children’s Research Hospital. Pediatric Therapy–related Myeloid Neoplasms (tMN): This study makes use of data generated by the St. Jude Children’s Research Hospital Pediatric therapy–related myeloid neoplasm (tMN) Study [51].

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
