# Peer review of "The Chimeric Antigen Receptor T Cell Target Claudin 6 Is a Marker for Early Organ-Specific Epithelial Progenitors and Is Expressed in Some Pediatric Solid Tumor Entities"

_cancers, 2025, doi:10.3390/cancers17060920_

Round 1

Reviewer 1 Report

Comments and Suggestions for Authors

1.       GD2 is not just a lipid, but is more specifically referred to as a ganglioside.

2.       It is somewhat surprising that rituximab is not mentioned among the highly successful immunotherapeutic approaches for pediatric patients with hematologic malignancies.

3.       CAR T and bispecific antibodies targeting CLDN6 are mentioned in the beginning, by not ADCs. Generally, although the interest of two authors employed at BioNTech sponsoring CLDN6-specific CAR T trial is clear, mentioning CAR T in the title sounds somewhat extraneous, as the study itself has nothing to do with CAR Ts. It is a thorough study of the expression pattern of CLDN6 in normal and cancerous tissues, but CLDN6 can be used as a target not only in the context of CAR T cells but also in monoclonals, ADCs, and BiTEs - as is appropriately mentioned in the Discussion. I see no point having "CAR T" mentioned in the title, as it attempts to "sell" the story higher than it merits.

4.       No need to capitalize the words lymphoma and leukemia.

5.       “Week intensity” should be changed to “weak intensity”.

6.       It is unclear why weak intensity CLDN6 staining was not considered/disregarded, as the authors are well aware of and appropriately reference the reports when weak expression of the target antigen on healthy organs/tissues has resulted in lethal outcomes and/or serious adverse events of CAR T cell therapy. Given that this information is already available, it should be made public for other researchers to consider.

7.       Figure 1: how come no ovarian tumors are shown, albeit those are present in the St.Jude cloud dataset (ovarian dysgerminoma n= 6, mixed germ cell tumor of the ovary, n=10)?

8.       Small intestin (table 1) to be replaced with "small intestine".

9.       Meaning of Sigma and Brustdrüse (mammary gland?) is unclear.

10.   ewing sarcoma to be spelled as "Ewing sarcoma".

11.   Rhabdoidtumor to be replaced by “rhabdoid tumor”, liversarcoma by “liver sarcoma”, pleuroplumonal by “pleuropulmonary”, Mesoblastic nephrom by “mesoblastic nephroma”

12.   Line 237: is 10[5] meant instead of 105? Same for line 370: 104.

13.   Figures 3 and 4: size bars and the font size used across the images should be made legible/uniform.

14.   In Table 4, it is unclear why most of the numbers have been rounded except for several numbers listed for postnatal development x Tubuli, Bowman’s capsule and Collecting Tubuli rows? % symbol can be safely removed from the table, as the figure legend clearly specifies that the percentage is shown.

15.   Supplemental Table 1: nephroblatoma to be replaced with "nephroblastoma", nuclear – remove umlaut, positiv – replace with “positive”, de-abbreviate “n.d.” in the legend, replace “yolc” with “yolk”, Dysgerminom with “dysgerminoma”, replace “w” with “f” in the “sex” column (female instead of woman), cytoplasma with “cytoplasm”, Rhabdoidtumor with “rhabdoid tumor”. How come the age is not known for several samples (ID12, 17)? What was meant by “mixed maligne” (ID 11)?

Author Response

We sincerely thank the reviewer for the numerous valuable suggestions. The lanes refer to the clean version (PDF)

1.GD2 is not just a lipid, but is more specifically referred to as a ganglioside.

The reviewer is right and we changed lipid to ganglioside (line 80)

  1.  It is somewhat surprising that rituximab is not mentioned among the highly successful immunotherapeutic approaches for pediatric patients with hematologic malignancies.

We thank the reviewer for this comment and have added Non-Hodgkin´s lymphoma  and CD20 (lane 79-80)

3. CAR T and bispecific antibodies targeting CLDN6 are mentioned in the beginning, by not ADCs. Generally, although the interest of two authors employed at BioNTech sponsoring CLDN6-specific CAR T trial is clear, mentioning CAR T in the title sounds somewhat extraneous, as the study itself has nothing to do with CAR Ts. It is a thorough study of the expression pattern of CLDN6 in normal and cancerous tissues, but CLDN6 can be used as a target not only in the context of CAR T cells but also in monoclonals, ADCs, and BiTEs - as is appropriately mentioned in the Discussion. I see no point having "CAR T" mentioned in the title, as it attempts to "sell" the story higher than it merits.

We highlighted CAR-T cells because of their notable success in treating pediatric tumors with poor prognosis. However, a significant concern is that the toxicity associated with CAR-T therapy can be particularly severe if the targeted antigens are also expressed in normal tissues. This is in contrast to monoclonal antibodies and antibody-drug conjugates (ADCs) targeting the same antigens, such as HER2, which may have a more favorable safety profile.

We clarified this point by adding “Immunotherapy using CAR-T cells targeting GD2 has shown remarkable success in treating pediatric tumors with poor prognosis, such as high-risk neuroblastoma and dif-fuse midline glioma [7,8]. However, translating this approach to other solid tumors re-mains difficult, partly due to the scarcity of optimal targets. A major concern is the poten-tial for CAR-T cells to cross-react with healthy tissues, leading to unacceptable toxicity from on-target/off-tumor effects [9]. Consequently, it is essential that the targeted antigens for CAR-T cells are not expressed on normal cells critical for survival. One such antigen candidate is Claudin 6 (CDN6). Lane 98-105

4.     No need to capitalize the words lymphoma and leukemia.

We have corrected it (lane 139)

5. “Week intensity” should be changed to “weak intensity”. We have corrected it (lane 158)

6. It is unclear why weak intensity CLDN6 staining was not considered/disregarded, as the authors are well aware of and appropriately reference the reports when weak expression of the target antigen on healthy organs/tissues has resulted in lethal outcomes and/or serious adverse events of CAR T cell therapy. Given that this information is already available, it should be made public for other researchers to consider.

We have included the definition of score +1 (lane 158-159). The scores, also +1, are listed in Table 3 (Organ specific expression of CLDN6 in normal pediatric tissues), Table 4 (Transient expression of CLDN6 in kidneys during normal development) and Supplemental table 1 (expression in tumor tissues).

7. Figure 1: how come no ovarian tumors are shown, albeit those are present in the St.Jude cloud dataset (ovarian dysgerminoma n= 6, mixed germ cell tumor of the ovary, n=10)?

We have included in the Figure the tumor samples of ovarian dysgerminoma (n=3) and mixed germ cell tumor of the ovary (n=3)  (some samples with FeatureCount Data are normal and were not included in the figure) and commented it at lane 181. The samples were not included in the former version because we used only entities with at least 5 tumor samples. With the inclusion of the two entities, the high expression of Cldn 6 is evident. We thank the reviewer for this suggestion.

8. Small intestin (table 1) to be replaced with "small intestine". We have changed it

9. Meaning of Sigma and Brustdrüse (mammary gland?) is unclear. Sigma was changed to sigmoid colon, brüstdruse was eliminatewd in Table 1

10. ewing sarcoma to be spelled as "Ewing sarcoma". We have changed it

11. Rhabdoidtumor to be replaced by “rhabdoid tumor”, liversarcoma by “liver sarcoma”, pleuroplumonal by “pleuropulmonary”, Mesoblastic nephrom by “mesoblastic nephroma” We have changed all

12. Line 237: is 10[5] meant instead of 105? Same for line 370: 104. Yes, we changed both

13. Figures 3 and 4: size bars and the font size used across the images should be made legible/uniform. We improved it in Figure 3, 4 and 5

14. In Table 4, it is unclear why most of the numbers have been rounded except for several numbers listed for postnatal development x Tubuli, Bowman’s capsule and Collecting Tubuli rows? % symbol can be safely removed from the table, as the figure legend clearly specifies that the percentage is shown.

We reorganised and simlify the table. The % of CLDN6 positive cells is now reported as mean and standard deviation

15. Supplemental Table 1: nephroblatoma to be replaced with "nephroblastoma", nuclear – remove umlaut, positiv – replace with “positive”, de-abbreviate “n.d.” in the legend, replace “yolc” with “yolk”, Dysgerminom with “dysgerminoma”, replace “w” with “f” in the “sex” column (female instead of woman), cytoplasma with “cytoplasm”, Rhabdoidtumor with “rhabdoid tumor”. How come the age is not known for several samples (ID12, 17)? What was meant by “mixed maligne” (ID 11)? We thank again the reviewer for the improvements and have done all the changement in the Table. We added one missing age.  Unfortunatly for the other sample the age was not recorded. We apologise for this.

Reviewer 2 Report

Comments and Suggestions for Authors

The manuscript is well-organized and uses strong methods. The study provides solid evidence that CLDN6 is a promising therapeutic target. The authors combined transcriptomic and immunohistochemistry data to analyze CLDN6 expression in pediatric tumors and normal tissues. However, they only examined CLDN6 as a marker and did not perform functional assays to test its role in tumor progression or treatment response. I believe adding some validation experiments would make the study more convincing.

Author Response

We thank the reviewer for the positive feedback. We agree that to show relevance of CLDN6 in tumor progression will support its importance as target. For this, we have included in the discussion more literature on its dual role in tumor progression and resistance (Lane 432-438 of the clean version (PDF)): "Moreover, CLDN6 supports cell migration and proliferation, can enhance chemoresistance and its expression is associated with the patient prognosis of a variety of tumors [28-30]. Targeting antigens with an important role in tumor’s viability limits the potential for cancer cells to develop escape variants. However, a dual role of CLDN6 in tumor progression has been also discussed and functional studies addressing the role of CLDN6 in pediatric tumors should be performed in the future [25]. "

Reviewer 3 Report

Comments and Suggestions for Authors

Excellent work.

It follows a previous Nature Medicine publication https://www.nature.com/articles/s41591-023-02612-0 with a focus on Pediatric individuals and patients.

I only have 2 suggestions:

Line 100: is the word "basket" correct?

Line 448: the following sentence

The EWS-WT1 transcription factor is the 448
driver of tumorigenesis in DSRCT and it acts by up-regulating the expression of several 449
growth factors. 450

Needs a reference.

Author Response

We thank the reviewer for the positive comments

Line 100: is the word "basket" correct?

Yes, this a type of clinical trial. In basket trials, patients who have different types of cancer all receive the same treatment that targets the specific mutation or biomarker found in their cancer.

Line 448: the following sentence

The EWS-WT1 transcription factor is the 448
driver of tumorigenesis in DSRCT and it acts by up-regulating the expression of several 449
growth factors. 450

Needs a reference.

We have added as reference: Desmoplastic small round cell tumor is dependent on the EWS-WT1 transcription factor. Gedminas JM, Chasse MH, McBrairty M, Beddows I, Kitchen-Goosen SM, Grohar PJ. Oncogenesis. 2020 Apr 28;9(4):41. doi: 10.1038/s41389-020-0224-1.PMID: 32345977

Reviewer 4 Report

Comments and Suggestions for Authors

Seidmann et al. provide valuable insights into the expression of Claudin 6 (CLDN6), an oncofetal membrane protein, and its potential as a therapeutic target for CAR-T cell-based immunotherapy in pediatric malignancies. This study systematically evaluates CLDN6 expression across pediatric solid tumors, including desmoplastic small round cell tumors (DSRCT), germ cell tumors (GCT), and Wilms tumors (WT), in comparison to its expression in normal tissues. The findings indicate that CLDN6 is highly expressed in select tumor types, particularly in GCTs and DSRCT, underscoring its relevance as a promising target for CAR-T cell therapy in pediatric oncology. The article is well-structured and informative. Below are some minor points the authors should address:

Minor Points:

  1. What are the molecular mechanisms governing the postnatal downregulation of CLDN6 expression in epithelial cells?
  2. What are the potential challenges associated with targeting CLDN6 in pediatric patients, particularly concerning off-target cytotoxicity and immunogenicity?
  3. What methodologies can be employed to quantitatively assess CLDN6 expression in tumors, and do expression levels vary across tumor stages or histological subtypes?

Author Response

We thanks the reviewer for the positive comments and discussed all the questions (the lanes refer to the clean version (PDF)

  1. What are the molecular mechanisms governing the postnatal downregulation of CLDN6 expression in epithelial cells?

Regulation of Claudin 6 is not well understood so far. However, promoter methylation plays a role and we have added it in the discussion at lane 426

2. What are the potential challenges associated with targeting CLDN6 in pediatric patients, particularly concerning off-target cytotoxicity and immunogenicity?

We have added at lane 564: 

In terms of expected toxicity, the immune system of pediatric patients may respond more vigorously or differently than that of adults to foreign proteins or engineered immune therapies, increasing the likelihood of immune-mediated adverse events. These events can include cytokine release syndrome (CRS) and immune effector cell–associated neurotoxicity syndrome (ICANS). Notably, CRS was observed in 46% of adult patients treated with CAR-T cells targeting CLDN6 [13]. Given the potential for such reactions, it is essential to implement supportive care measures to manage and mitigate these adverse effects effectively. Finally, the high degree of structural and sequence similarity within members of the CLDN family raises the question of possible off-target toxicities, that occur when the transduced T-cell population unexpectedly attacks an antigen other than the intended. Notably, the extracellular regions of CLDN6 and CLDN9 vary by just three amino acids and several antibodies failed to show selective binding to CLDN6 [46]. The anti-CLDN6 CAR T cells identify the antigen using a single-chain variable fragment derived from the monoclonal antibody IMAB206-C46S. Although CLDN9 is expressed in healthy tissues such as the pituitary gland, no off-target toxicity has been reported, suggesting the antibody's specificity [13,47].

3. What methodologies can be employed to quantitatively assess CLDN6 expression in tumors, and do expression levels vary across tumor stages or histological subtypes?

Immunohistochemistry is the best method to assess Claudin 6 expression. The method is only semi- quantitative, but our results, particularly in Wilms tumors, indicate that the expression is heterogeneous and therefore methods such qRT-PCR  from bulk tissues are not sufficient. Single cells transcriptomic and/or flow cytometry could be implemented  to address the amount of Cldn6, the heterogeneity and the co-expression with other markers. We have added in the discussion:

“Flow cytometry and single-cell transcriptomics could be employed to investigate the heterogeneity of CLDN6 expression, enabling quantification and analysis of its co-expression with other markers [43]. These approaches could provide new insights into the variations in CLDN6 levels across different cell populations.” Lane 541-544

Concerning association with the stage or histological subtypes we don’t have enough samples to perform the analysis and the information was not present in the expression data we have used in Figure 1. We have discussed this at lane 527-531:

“Due to the limited number of samples in our cohort, we were unable to analyze the correlation between CLDN6 expression and tumor stage or histological subtypes. However, a previous study has demonstrated significant correlations between CLDN6 expression, molecular subtypes, and clinical characteristics in various adult cancers, and particularly in endometrial carcinoma, using transcriptomic data [30]